

# Comparative transcriptomics characterized the distinct biosynthetic abilities of terpenoid and paeoniflorin biosynthesis in herbaceous peony strains

Baowei Lu[1], Fengxia An[1], Liangjing Cao[2], Qian Gao[2], Xuan Wang[3], Yongjian Yang[1], Pengming Liu[1], Baoliang Yang[1], Tong Chen[4], Xin-Chang Li[2], Qinghua Chen[1] and Jun Liu[2]

[1] Bozhou University, Bozhou, Anhui, China
[2] National Key Facility for Crop Resources and Genetic Improvement, Institute of Crop Sciences, Chinese Academy of Agricultural Sciences, Beijing, China
[3] Department of Biological Sciences, Xian Jiaotong-Liverpool University, Suzhou, China
[4] National Resource Center for Chinese Materia Medica, Academy of Chinese Medical Sciences, Beijing, China

Corresponding authors
Fengxia An, An_fengxia@163.com
Jun Liu, liujun@caas.cn

## ABSTRACT

The herbaceous peony (*Paeonia lactiflora* Pall.) is a perennial flowering plant of the Paeoniaceae species that is widely cultivated for medical and ornamental uses. The monoterpene glucoside paeoniflorin and its derivatives are the active compounds of the *P. lactiflora* roots. However, the gene regulation pathways associated with monoterpene and paeoniflorin biosynthesis in *P. lactiflora* are still unclear. Here, we selected three genotypes of *P. lactiflora* with distinct morphologic features and chemical compositions that were a result of long-term reproductive isolation. We performed an RNA-sequencing experiment to profile the transcriptome changes of the shoots and roots. Using de novo assembly analysis, we identified 36,264 unigenes, including 521 genes responsible for encoding transcription factors. We also identified 28,925 unigenes that were differentially expressed in different organs and/or genotypes. Pathway enrichment analysis showed that the *P. lactiflora* unigenes were significantly overrepresented in several secondary metabolite biosynthesis pathways. We identified and profiled 33 genes responsible for encoding the enzymescontrolling the major catalytic reactions in the terpenoid backbone and in monoterpenoid biosynthesis. Our study identified the candidate genes in the terpenoid biosynthesis pathways, providing useful information for metabolic engineering of *P. lactiflora* intended for pharmaceutical uses and facilitating the development of strategies to improve marker-assist *P. lactiflora* in the future.

## INTRODUCTION

The herbaceous peony (*Paeonia lactiflora* Pall.) is a flowering plant in the family Paeoniaceae, which is native to Central and eastern Asia (*Zhao et al., 2018*, *2017*). Its dried
root is harvested without the bark in the autumn from plants that are between 3 and 5 years of age; this harvested material is named Radix Paeoniae Alba or Baishao and is a well-known Chinese herb, used for over 2,000 years (*He & Dai, 2011*; *Zha, Cheng & Peng, 2012*). A water/ethanol extract of Radix Paeoniae Alba, now known as Total Glucosides of Peony (TGP), was originally used in the treatment of typhoid fever (*Li, Chen & Shen, 2011*). Subsequently, TGP has been widely prescribed for fever, rheumatoid arthritis, hepatitis, muscle cramping and spasms, systemic lupus erythematosus, and dysmenorrhea (*Fan et al., 2012*; *He & Dai, 2011*; *Ji et al., 2013*; *Mao et al., 2012*; *Nam et al., 2013*).

Paeoniflorin (C23H28O11, molecular weight = 480.45) is the major medicinal component in *P. lactiflora* roots. In vitro and in vivo studies in animal models have confirmed that TGP, paeoniflorin, benzoylpaeoniflorin, galloylpaeoniflorin and their derivatives, are medicinally active compounds with multiple pharmacological effects (*Fan et al., 2012*; *He & Dai, 2011*; *Zhou & Wink, 2018*). TGP can inhibit acute and subacute inflammation, an effect which is potentially mediated by the suppression of prostaglandin E2, leukotriene B4, and nitric oxide, as well as the intracellular calcium ion concentration (*He & Dai, 2011*; *Xu et al., 2016*). TGP has been known to protect cells against $Ca^{2+}$ overload and oxidative stress (*Zhang et al., 2017*). Moreover, the components of TGP, as important immunomodulatory effectors, can regulate the proliferation and apoptosis of lymphocytes and balance the production of proinflammatory cytokines in a dose-dependent manner (*He & Dai, 2011*; *Hu et al., 2018*). In addition, paeoniflorin and its derivatives were shown to inhibit tumor growth and macrophage-mediated lung metastases (*Ou et al., 2011*; *Wu et al., 2015*).

Paeoniflorin is a monoterpene glucoside that is biosynthesized from geranyl-pyrophosphate (GPP). GPP is produced via a conversion from the universal terpenoid precursor, Isopentenyl pyrophosphate (IPP). In plants and bacteria, IPP is produced from the two terpene biosynthesis pathways, the mevalonate pathway (MVA), and the 1-deoxy-D-xylulose-5-phosphate/methyl-erythritol-4-phosphate (MEP/DOXP) pathway (Fig. 1) (*Kanehisa et al., 2012*; *Ren et al., 2009*; *Xie et al., 2011*). The MVA pathway reactions take place in the cytosol and are catalyzed by enzymes including hydroxyl methylglutaryl-CoA synthase (HMGCS), acetyl-CoA *C*-acetyltransferase (AACT), HMG-CoA reductase (HMGCR), mevalonate kinase (MK), and phosphomevalonate kinase (PMK). The DXP/MEP pathway is catalyzed in the plastids. Pyruvate and glyceraldehyde 3-phosphate are converted by 1-deoxy-D-xylulose-5-phosphate synthase (DXS) and 1-deoxy-D-xylulose-5-phosphate reductoisomerase (DXR) to 1-deoxy-D-xylulose 5-phosphate and 2-C-methyl-D-erythritol 4-phosphate, respectively. The products are subsequently catalyzed by 4-diphosphocytidyl-2-C-methyl-D-erythritol synthase (ISPD), CDP-ME kinase (CDPME), and 2-C-methyl-D-erythritol 2,4-cyclodiphosphate synthase (ISPF) to mediate the formation of 2-C-methyl-D-erythritol 2,4-cyclopyrophosphate, which is then converted to (E)-4-hydroxy-3-methyl-but-2-enyl pyrophosphate (HMB-PP) by HMB-PP synthase (HDS). HMB-PP is converted to IPP and dimethylallyl pyrophosphate (DMAPP) by HMB-PP reductase (ISPH). IPP and DMAPP are condensed by geranyl pyrophosphate synthase (GPS) to produce GPP. In addition to producing a monoterpene, GPP is also a precursor to sesquiterpenes and

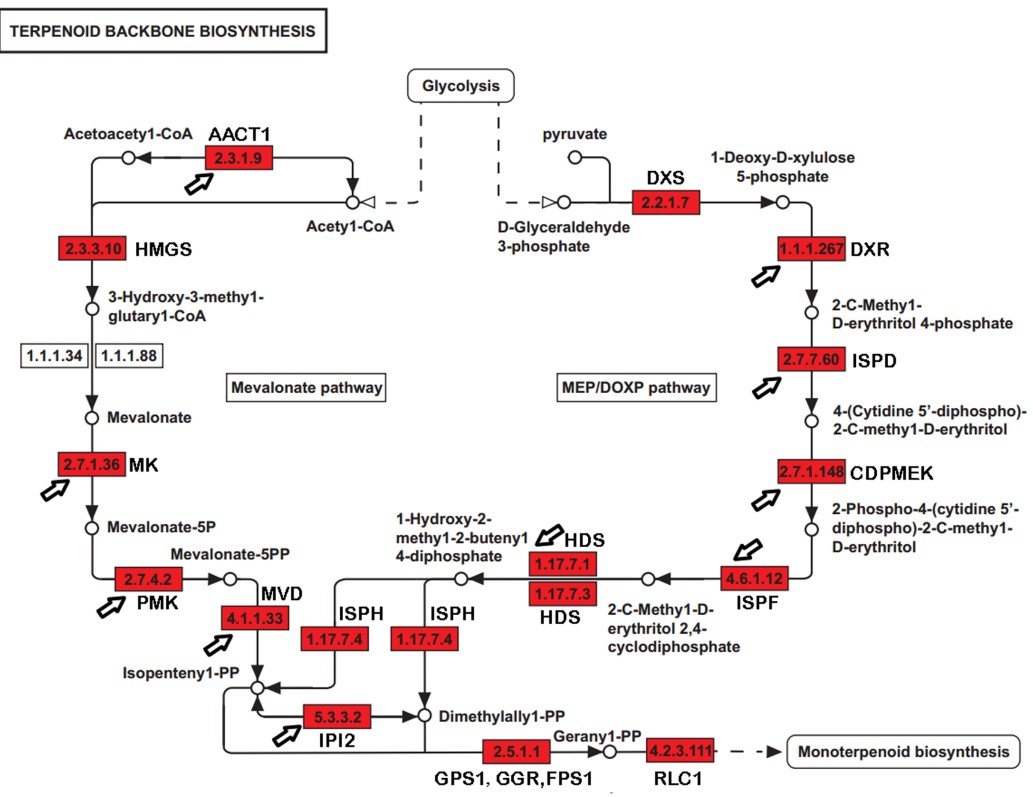

**Figure 1 The genes encoding enzymes in terpenoid backbone and monoterpenoid biosynthesis pathways in *P. lactiflora*.** The identified *P. lactiflora* enzymes are highlighted by red rectangles. The newly identified enzymes compared with the previous study are indicated by arrows (*Yuan et al., 2013*).

diterpenes. The conversion from GPP to alpha-terpineol is critical for producing the monoterpene, which is catalyzed by (−)-alpha-terpineolsynthase (RLC1). Paeoniflorin can be modified by benzoic acid and gallic acid to produce benzoylpaeoniflorin and galloylpaeoniflorin, respectively. Benzoic acid and gallic acid are catalyzed by 3-deoxy-7-phosphoheptulonate synthase, 3-dehydroquinate synthase, and 3-dehydroquinatedehydratase/shikimate dehydrogenase.

With a long history of domestication and selection, the *P. lactiflora* strains used for medical purposes contain high levels of paeoniflorin and are nearly completely infertile due to embryo abortion in their traditional planting regions, like the Bozhou area. Strains have been reproduced through the vegetative propagation of shoots for hundreds of years. Thus, these *P. lactiflora* accessions are reproductively isolated and may serve as suitable resources in the investigation of the genetic and molecular basis of the paeoniflorin biosynthesis pathways. Using sequence homology to search for the known sequences and domains, several studies identified a group of genes involved in paeoniflorin biosynthesis. For example, a previous study identified 24 genes, including eight with full-length cDNA sequences and revealed transcriptional and phylogenetic associations with paeoniflorin biosynthesis (Fig. 1) (*Yuan et al., 2013*). However, the genes in the

paeoniflorin biosynthesis pathways and their expression patterns have not been profiled in *P. lactiflora* strains on a genome-wide basis.

High throughput sequencing technologies have revolutionized genomic and transcriptomic studies. Improved algorithms are now available for de novo reassembly of the transcriptome of a non-model plant species without a valid reference genome sequence (*Luo et al., 2017b*). In this study, we assembled the transcriptome of roots and shoots derived from the three strains of *P. lactiflora* using de novo assembly analysis. By aligning the assembled genes with public databases, we globally annotated 34,203 unigenes in *P. lactiflora*. For instance, our analysis identified 521 transcription factor genes. Moreover, we profiled gene expression levels and identified a group of tissue- and/or strain-differential expressed genes using differential expression analysis. From the list of differentially expressed genes among shoots vs. roots, we used homology based annotation with previously known gene families to identify the genes not yet identified from the known pathway. We verified the expression pattern of a selective group of candidate genes using the qRT-PCR assay. Our study provides a valuable dataset for updating our understanding of the gene regulatory network underlying paeoniflorin biosynthesis in *P. lactiflora*.

## MATERIALS AND METHODS

### Plant materials for RNA-Seq and HPLC

The *P. lactiflora* Pu–Bang, Xian–Tiao, and Guan–Shang strains were conserved and cultivated under field conditions at Bozhou University, Bozhou, China. The shoots and roots of 3-year old plants were isolated. To avoid circadian effects, we harvested all the tissues in the afternoon of the same day. The samples for RNA-seq with three biological replicates were frozen in liquid nitrogen immediately after harvesting. The isolated samples and purified RNA were stored at −80 °C. To measure paeoniflorin level by HPLC, roots of 3-year old plants were dried after removing its barks.

### RNA extraction, library construction, and Illumina sequencing

The total RNA of individual samples was extracted and purified with the RNeasy® Plant Mini Kit (QIAGEN, Hilden, Germany). RNA concentration was measured using a Nanodrop 2100 spectrophotometer. RNA Integrity values were checked using an Agilent Bioanalyzer. The samples with a RIN score >8.5 were used for library construction (*Liu et al., 2014*). The sequencing libraries were generated using a NEB Next Ultra RNA Library Prep Kit for Illumina (New England Biosystems, Waltham, MA, USA), following the manufacturer's recommendations. Library sequencing was performed on a Hiseq X10 system with 150-cycle paired-end sequencing protocol (Illumina, San Diego, CA, USA).

### Bioinformatics analysis of RNA-seq datasets

RNA-seq datasets were checked using FastQC (*Brown, Pirrung & McCue, 2017*). Referring the methods used in recent studies with modification (*Bedre et al., 2016*; *Lu et al., 2018*), we assembled the transcriptome using Trinity (*Haas et al., 2013*). We aligned read

sequences using HISAT2 (*Kim, Langmead & Salzberg, 2015*). Read count of each gene was called using HTseq-count (*Anders, Pyl & Huber, 2015*). Fragments per kilobase of exon per million fragments mapped of assembled transcripts (FPKM) were calculated and normalized using DESeq2 with global normalization parameters (*Anders, Pyl & Huber, 2015*; *Love, Huber & Anders, 2014*; *Quinn & Chang, 2016*; *Zhang et al., 2014*). The sequences of the assembled unigenes were annotated by Trinotate (*Haas et al., 2013*). Coding regions of unigenes were predicted using Transdecoder (*Haas et al., 2013*). BLAST v2.7.1 was performed to determine the sequence homology (*e*-value cutoff of $1e^{-5}$) to UniProt/Swiss-Prot, HMMR v3.1b2, EggNOG v4.5.1, and metabolic pathways were analyzed using KEGG database (*Kanehisa et al., 2016*). For the unigene with annotated homolog, we used the criterion of FPKM more than 0.8 to filter out the lowly expressed genes; for those without homolog, we used FPKM more than 1 as the cut-off criterion. Differential expression analysis was carried out using DESeq2 (*Anders & Huber, 2010*). Genes with normalized fold-change greater than 2, significance *P*-value less than 0.05, and Benjamini–Hochberg false discovery rate less than 0.1 were considered to be differentially expressed genes.

## Quantitative detection of paeoniflorin

We referred to the previous method in order to measure paeoniflorin in samples (*Yuan et al., 2013*). The dried samples (0.50 g) were weighed and extracted with 50 mL of 50% aqueous methanol with ultrasonication for 30 min. The extracted samples were diluted with 50 mL 50% aqueous methanol and filtered with a 0.45-µm Millipore filter membrane (Millipore, MA, USA) at 25 °C. We used the Agilent 1200 LC Series (Agilent Technologies, Palo Alto, CA, USA) High Performance Liquid Chromatography (HPLC) system to measure the paeoniflorin abundance. The wavelength was set at 230 nm with a flow rate of 1.0 mL/min at a temperature of 25 °C. Standard compounds were purchased from the National Institutes for Food and Drug Control and the linearity of the standard compounds was checked at seven concentration solutions.

## Quantitative RT-PCR

A total of 1 to 2 µg RNA samples were treated by DNase I (RNeasy plant mini kit) and were reverse transcribed with oligo (dT) primer and SuperScript III (Invitrogen, Carlsbad, CS, USA). cDNA samples were analyzed using quantitative PCR with SYBR Premix Ex Taq (Takara, Shiga, Japan) and a Biorad CFX96 real-time PCR system. The *P. lactiflora* Actin transcript sequence (JN105299) was used as endogenous reference genes to design primers to normalize the expression levels among samples (*Qi et al., 2018*; *Yuan et al., 2013*). The qRT-PCR reactions were carried out with two-step cycles (5 s 95 °C denaturation, 30 s 60 °C annealing and extension) and 45 cycles of amplification (Fig. S1A). The melting curves of primers were checked to ensure the primer efficiencies (Fig. S1B–S1L). We used three technical replicates to produce the average expression levels of the genes relative to that of the reference gene using the $2^{-\Delta\Delta CT}$ method (*Livak & Schmittgen, 2001*). The primers are listed in Table S1.

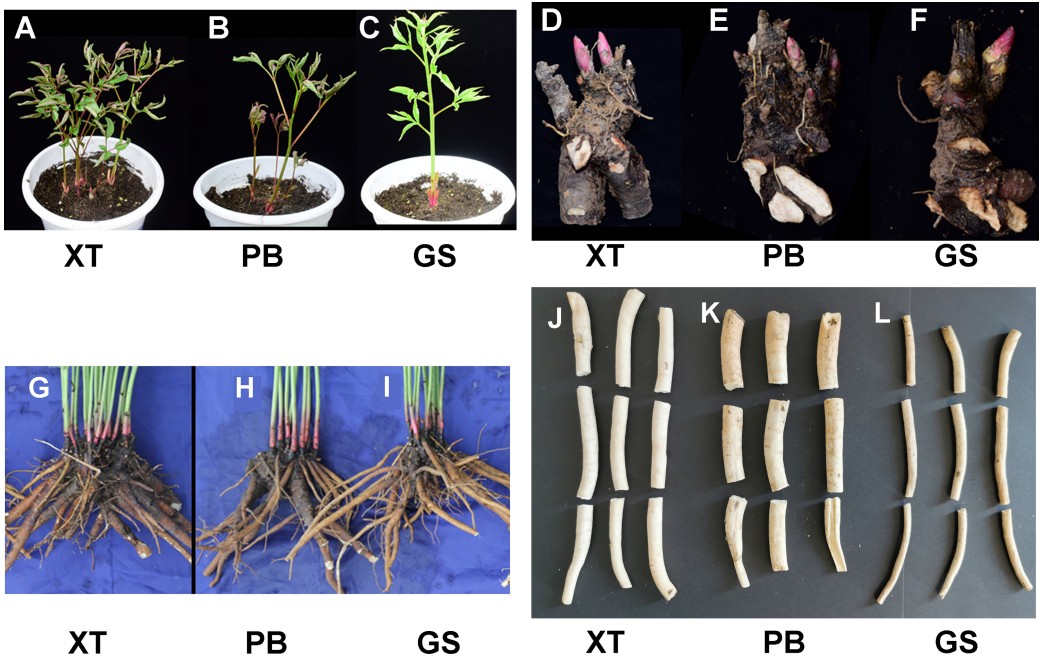

**Figure 2** **The developmental features of the three *P. lactiflora* strains.** (A–C) The 30-day-old plants of Xian–Tiao (XT), Pu–Bang (PB), and Guan–Shang (GS) strains, which were grown from the shoots (D–F) isolated from 3-year-old plants. (G–I) and (J–L) show fresh roots and dried roots without the bark of 3 year-old plants, respectively.

## RESULTS

### RNA-Seq and de novo assembly of the *P. lactiflora* transcriptome

The *P. lactiflora* Pu–Bang (PB) and Xian–Tiao (XT) accessions are the most widely used herbaceous stains for medical uses due to their high levels of paeoniflorin, which is derived mainly from roots of 3-year old plants; whereas the Guan–Shang (GS) accession contains less paeoniflorin and is usually cultivated for ornamental uses. The morphological features of the 1- and 3-year old plants of the three strains were shown in Fig. 2, respectively. We determined the paeoniflorin levels of the isolated shoot and root samples of 3-year old plants using High Performance Liquid Chromatography (HPLC). The results confirmed the accumulated levels of paeoniflorin in PB and XT roots compared with those of GS (Fig. 3A).

To systematically identify genes and explore the gene expression network underlying paeoniflorin biosynthesis in *P. lactiflora*, we purified the RNA samples with three biological replicates derived from the shoots and roots of 3-year old PB, XT, and GS plants, and carried out the RNA-sequencing analysis using the Illumina paired-end 150 bp protocol. After filtering out the low quality reads, we obtained 775.73 million reads in total (Table S2). Using Trinity (*Haas et al., 2013*), we performed de novo transcriptome assembly and obtained 36,264 unigenes encoding 72,910 transcripts with 986 nt contig N50 length and 42.7% average GC content (Table 1). We also checked the completeness of

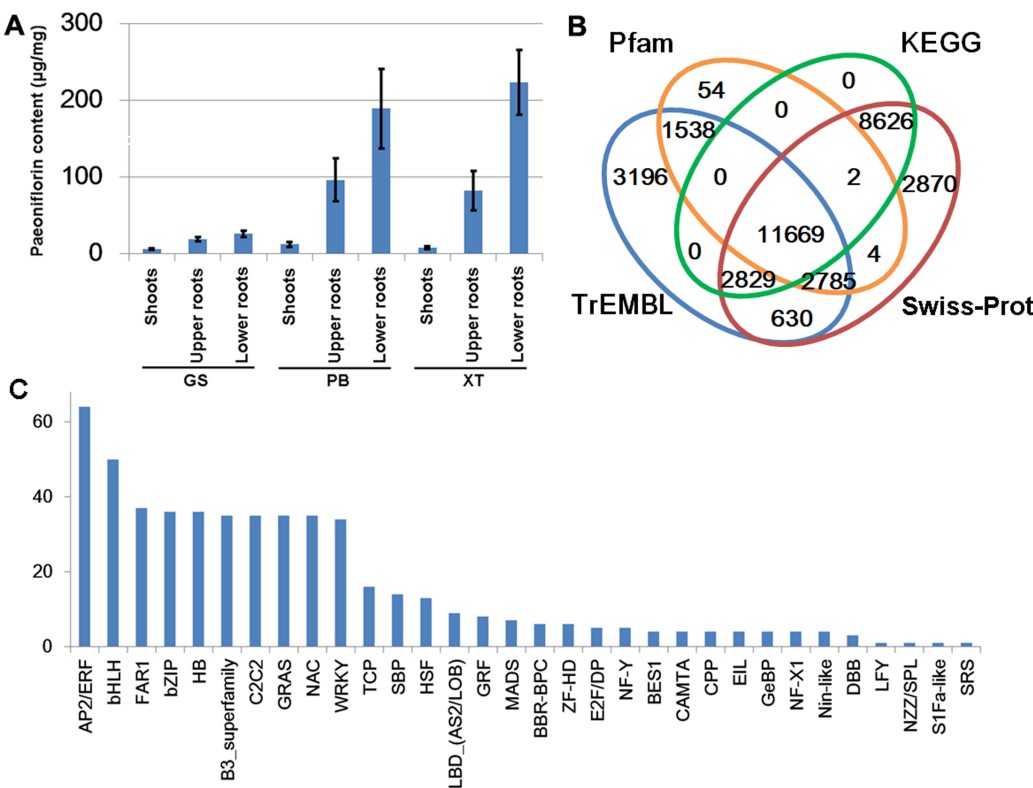

**Figure 3 Paeoniflorin levels and gene annotation in the three *P. lactiflora* accessions.** (A) Paeoniflorin levels measured by HPLC system. Bars show standard deviations of five biological replicates. (B) Functional annotation of unigenes using the four databases. (C) Transcription factor genes identified in *P. lactiflora*.

**Table 1 Transcriptome assembly of *P. lactiflora* unigenes.**

| Item | Value |
| --- | --- |
| Total unigenes | 36,264 |
| Total trinity transcripts | 72,910 |
| GC content (%) | 42.7 |
| Contig N10 (nt) | 1,901 |
| Contig N20 (nt) | 1,471 |
| Contig N30 (nt) | 1,195 |
| Contig N40 (nt) | 986 |
| Contig N50 (nt) | 812 |
| Median contig length (nt) | 481 |
| Average contig length (nt) | 640.8 |
| Total assembled bases | 46,721,207 |

de novo assembled Unigenes by aligning the sequences with Swiss-Prot database (*The UniProt Consortium, 2017*). The results showed that 7,131 (50%) out of the 14,165 aligned sequences have more than 40% coverage of known transcripts (Tables S3 and S4).

## Functional annotation of expressed genes in the three *P. lactiflora* accessions

Due to embryo abortion and vegetative propagation, the *P. lactiflora* accessions have been undergoing reproductive isolation with a long cultivation history in the Bozhou region and were thought to be genetically distinct from each other (*Zhou et al., 2002*). However, the genomic evidence supporting this point is still lacking. We measured the expression levels of unigenes by calculating normalized Fragments Per Kilobase of exon per million fragments Mapped (FPKM). The unigenes with an FPKM value higher than 1 in at least one were used to perform hierarchical clustering analysis based on the Pearson correlation efficiency. We analyzed the hierarchical structure of the gene expression levels on a genome-wide basis (Fig. S2A). Most of the biological replicate samples belonged to the same clusters. Principal component analysis results also showed a similar result confirming a high level of reproducibility of the biological replicate samples (Fig. S2B). However, the tissues and strains were distributed in different cluster clades. It was noted that all the clades derived from PB and XT were separated from those of GS, indicating that the strains for medicinal uses are genetically divergent from the strains used for ornamental purposes, possibly due to the selection and reproductive isolation among the strains.

We predicted the protein-coding potential for the unigenes using the Transdecoder and searched for the annotation for the unigenes by aligning the assembled transcripts and predicted peptide sequences to the protein sequences annotated by Swiss-Prot, TrEMBL, Pfam and KEGG databases using Trinotate, BLASTP and BLASTX (*Boutet et al., 2016*; *Camon et al., 2003*; *El-Gebali et al., 2019*; *Haas et al., 2013*; *Kanehisa et al., 2017*). In total, we identified 34,203 unigenes containing significant matches to the annotated genes/ proteins in at least one database (Fig. 3B). Of them, 28,083 (82%) were reproducibly detected by at least 2 data resources. The annotation information, predicted protein sequences, and FASTA-formatted sequences of these genes were provided in Supplemental Materials that could serve as a reference annotation for future studies (Table S3).

Transcription factors (TFs) with DNA-binding domains are the major regulators controlling the activity and specificity of the gene transcription process. We predicted genes encoding TFs in our assembled unigene dataset and identified 521 TF-encoding unigenes belonging to 32 TF families (Fig. 3C). Of these, AP2/ERF, bHLH, FAR1, bZIP and HB are the most abundant TF genes. The detailed information of the TF genes was provided in Table S5.

## Identification of tissue- and/or strain-differentially expressed unigenes

Next, we searched for the differentially expressed unigenes. For each strain, we compared shoots and roots and extracted lists of root/shoot specific upregulated genes (fold change of expression level >2 and false discovery rate <0.05). Our analysis identified 10,125 up-regulated unigenes in roots and 11,911 in shoots. There were 1,906–3,737 unigenes specifically up-regulated in the roots and/or shoots of each strain; whereas only 332 (3%) and 886 (7%) unigenes were up-regulated in the roots and shoots of all the three *P. lactiflora* strains, respectively (Fig. 4). This result showed that a large number of the
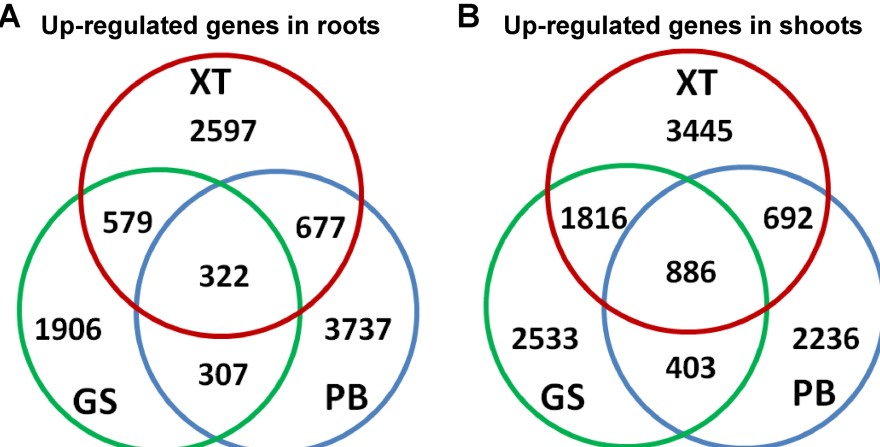

**Figure 4** **Identification of the differentially expressed genes.** The differentially expressed genes in roots (A) and shoots (B) of the three *P. lactiflora* accessions.

tissue-specific genes were also strain-specific, which is consistent with the fact that the three *P. lactiflora* strains have been undergoing genetic separation and selection during the last several centuries.

We performed the correlation analysis between the paeoniflorin levels and the expression levels of all the genes in shoots and roots of the three strains (Table S3). Our analysis identified 3,952 genes with the Pearson correlation coefficients higher than 0.6. Among the 161 gene with Pearson correlation coefficients higher than 0.9, we found several genes encoding transcription factors, E3 ubiquitin-protein ligase and oxidation-reduction related process, such as GT-2, ABCF4, SKP2A, FREE, GOR, TLP, RMA1H1, HAT5, XBAT31 and DELLA, suggesting transcriptional and post-translational regulation may be highly associated with paeoniflorin accumulation.

## Identification of genes in terpenoid and paeoniflorin biosynthesis pathway

To further dissect the regulation pathways of the unigenes, we analyzed the gene list enrichment using the Kyoto Encyclopedia of Genes and Genomes (KEGG) datasets and KOBAS3.0 with hypergeometric testing and Benjamini and Hochberg correction (*Kanehisa et al., 2012*; *Xie et al., 2011*). In total, we identified 71 significantly enriched KEGG pathways in *P. lactiflora* (Table S6). The list included several KEGG terms of secondary metabolite biosynthesis, such as: Terpenoid backbone biosynthesis (*P*-value < 0.0152), Glycerophospholipid metabolism (*P*-value < 0.0001), Inositol phosphate metabolism (*P*-value < 0.0001), Pyruvate metabolism (*P*-value < 0.0002), Seleno compound metabolism (*P*-value < 0.0298), Ascorbate and aldarate metabolism (*P*-value < 0.0172) and Butanoate metabolism (*P*-value < 0.0352). *P. lactiflora* are generally known to have abundant secondary metabolites (*Li et al., 2016a*; *Liu et al., 2017*; *Ma et al., 2016*). Our transcriptome and pathway enrichment results are consistent with the metabolism profiling studies.

The previous studies have identified 19 EST sequences in the terpenoid backbone biosynthesis pathways in *P. lactiflora*, including 7 with full-length cDNA sequences

(*Yuan et al., 2013*). However, the genes in the terpenoid backbone biosynthesis pathway have not been globally profiled and the enzyme catalyzing the initiation step from GPP to monoterpenoid biosynthesis has not been identified in *P. lactiflora* yet. In our datasets, we identified 32 genes with full-length CDSs encoding the enzymes controlling the major catalytic reactions in MVA and MEP pathways (Table S7). Compared with the previous study (*Yuan et al., 2013*), the genes encoding 10 previously reported enzymes were also identified in our study (Fig. 1). Moreover, we identified the unigene (E_H33980_c1_g4, *RLC1*) encoding (−)-alpha-terpineol synthase (EC 4.2.3.111) that can catalyze the conversion from GPP to alpha-terpineol (*Kulkarni et al., 2013*), a monoterpene precursor of paeoniflorin. Our analysis identified the genes encoding the enzymes that almost completely catalyzed the reactions from the glycolysis products to the terpenoid backbone and monoterpenoid biosynthesis in *P. lactiflora*.

The high accumulation of paeoniflorin in XT and/or BP roots suggests some genes in the paeoniflorin biosynthesis pathway may be highly expressed in XT and/or PB roots. We analyzed the expression patterns of the aforementioned 33 unigenes (Fig. 5). In XT roots, the expression levels of *AACT1*, *HMGS*, *MK*, *PMK*, *GGR*, *FPS1* were higher than those in GS roots; whereas the expression levels of *PMK*, *MVD*, *GGR* in PB roots were higher than those in GS roots. The other genes did not clearly reveal XT- and/or PB-root differential expression patterns. Note that XT and PB presented the distinct expression pattern of these genes, which was consistent with the fact that the two strains were genetically isolated with a long history (*Zhou et al., 2002*). To confirm the expression levels determined by the RNA-seq experiment, we used a quantitative Real-time PCR (qRT-PCR) assay to measure the expression levels of several paeoniflorin biosynthesis pathway genes and transcription factor genes (Fig. 6; Fig. S1). The expression levels of the transcription factor gene *BHLH94* showed a highly correlation with paeoniflorin levels. The paeoniflorin biosynthesis pathway gene *FPS1*, *AACT1*, *HMGS*, *IPI2* and *MK* as well as the transcription factor gene *DREB1D* and *NAC098* presented root- and/or strain-preferential expression patterns. Our qRT-PCR verification results were highly consistent with the RNA-seq analyses (Fig. 6; Fig. S3).

## DISCUSSION

The herbaceous peony has a large and complex genome. It was estimated that the genome size of the tree peony, a closely related species of *P. lactiflora*, is about 12.5 Gbp. Our analysis showed that many gene families have multiple copies also suggesting the chromosome duplication and/or gene family expansion event(s) in *P. lactiflora* genome. Thus, assembling the high-quality genome of *P. lactiflora* remains a challenge. Recently, transcriptome studies on a large number of plant species with complex genomes, such as smooth cordgrass (*Spartina alterniflora*) (*Bedre et al., 2016*), buckwheat (*Lu et al., 2018*), and *Caraganakor shinskii* (*Li et al., 2016b*), have been carried out. Transcriptome assembly is currently a feasible and cost-efficient technology to globally identify genes in the *P. lactiflora* genome because genome information is unavailable (*Luo et al., 2017a*, *2017b*). In our study, we identified 36,264 unigenes. Using the homolog alignment analysis, we identified a large number of these assembled unigenes encoded in known regulatory

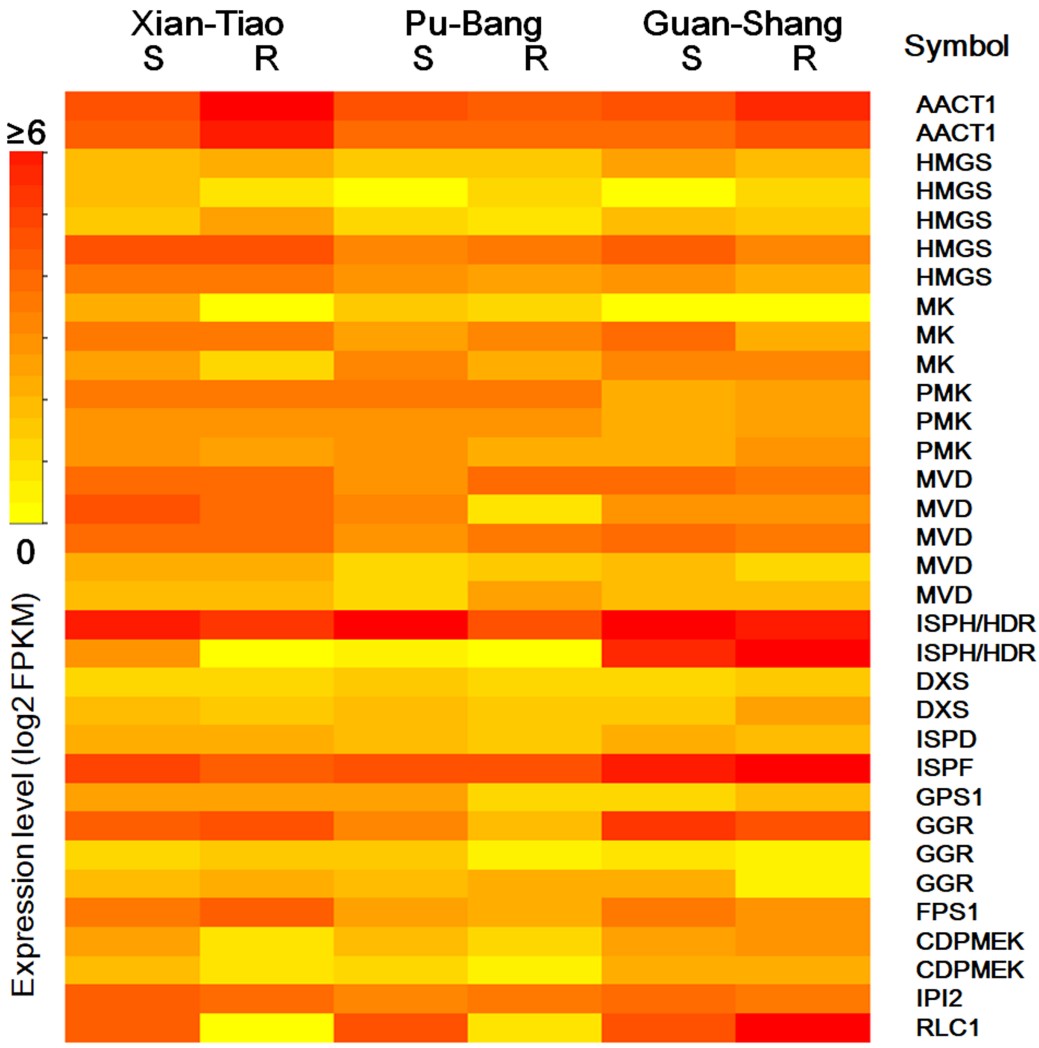

**Figure 5 The expression levels of the genes encoding enzymes in terpenoid backbone and monoterpenoid biosynthesis pathways in *P. lactiflora*.** The heatmap plot presents the mean value of expression levels of the three biological replicates. The symbols are given according to their expression levels. S and R indicate shoots and roots, respectively.

domains/motifs, suggesting their molecular functions. Of these, we identified 521 transcription factor genes belonging to 32 families and profiled their expression patterns. Our study identified a large number of previously un-annotated genes in the *P. lactiflora* genome, which could supply valuable molecular information for future functional studies.

Plant terpenoids are widely used as traditional herbal remedies and for their aromatic qualities. Terpenoids are highly abundant in several accessions of *P. lactiflora*, suggesting a high potential for terpenoid biosynthesis in them. However, the terpenoid biosynthetic pathways are not yet fully understood in *P. lactiflora*. Previous studies have identified the 19 EST sequences in the terpenoid backbone biosynthesis pathway in *P. lactiflora* (*Yuan et al., 2013*). Paeoniflorin is accumulating in roots of 3-year-old plants and lacking in leaves and its accumulation levels are highly differed in different strains (*Li, Chen & Shen, 2011*; *Yuan et al., 2013*). The tissue- and strain-specific accumulation pattern of

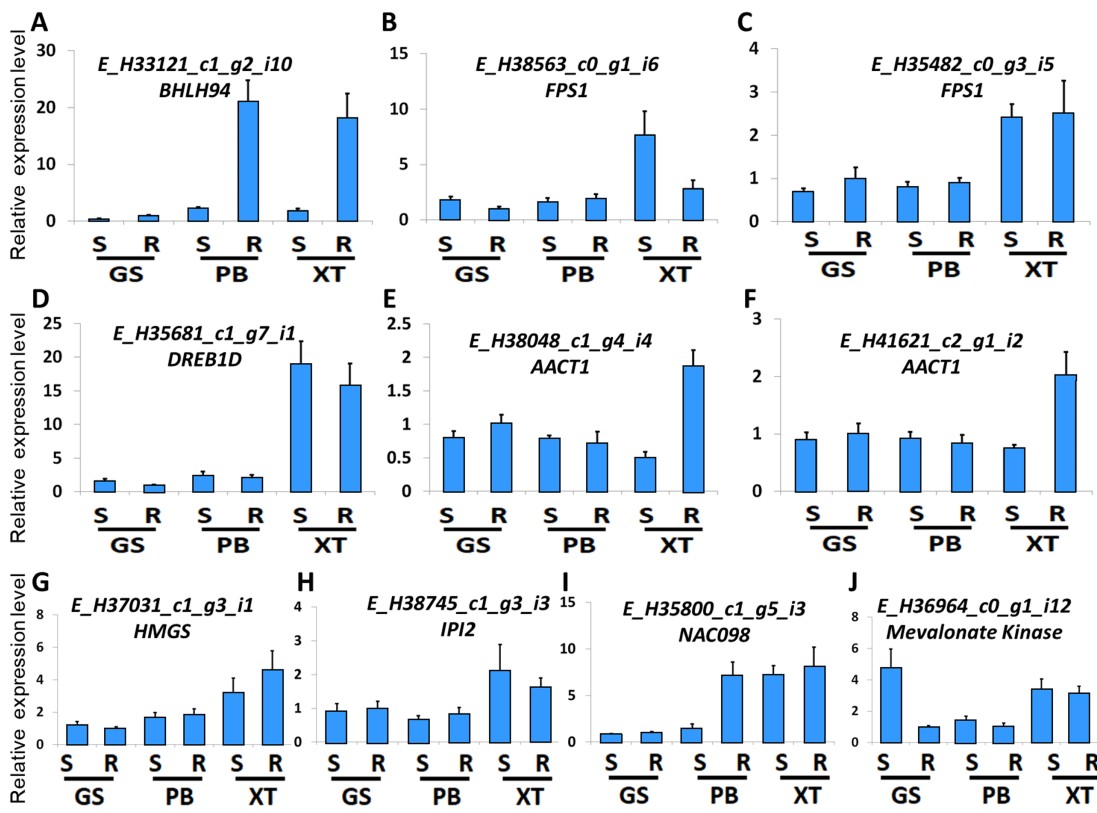

**Figure 6 Expression levels of the genes verified by qRT-PCR.** (A–J) The relative expression levels of the unigenes encoding BHLH94, FPS1, FPS1, DREB1D, AACT1, AACT1, HMGS, IPI2, NAC098, MK, respectively. The expression levels of GS root samples were used for normalization of the relative expression levels. Bars give standard errors of biological replicates ($n = 3$). S and R indicate shoots and roots, respectively.

paeoniflorin facilitates us to explore the molecular basis of paeoniflorin biosynthesis pathways. In our study, we carry out a transcriptome analysis and identify 32 unigenes in the pathway and profiled their expression patterns that covered most of the reactions in the terpenoid backbone biosynthesis pathway. The expression levels of *AACT1* showed specific accumulation in XT/PB roots; whereas most of the genes in the terpene backbone biosynthesis showed differential expression patterns between strains rather between tissues. This result raised two testable hypotheses: (1) The several genes with *AACT1* expression patterns are master genes in the pathway; (2) The root-specific expression patterns of these genes associated with paeoniflorin levels could be found in other strains. The gene sequences and their expression patterns identified in our study could serve as a reference dataset for follow up studies to test their expression specificities in more *P. lactiflora* strains associated with paeoniflorin levels. With this data, the other unidentified gene members and the complete terpenoid backbone biosynthesis pathway in *P. lactiflora* strains can be deciphered in the future, using homologous cloning, phylogenetic analysis, and catalytic kinetics experiments.

Our transcriptome profiling analysis uncovered the differential expression patterns of roots vs. shoots of three *P. lactiflora*. strains, with different paeoniflorin contents in the

roots, which is the producing tissue and had been determined higher in two strains. Our results suggested the genes in the terpenoid biosynthesis pathway were diversified during the selection process, which could supply valuable molecular information for future functional studies in *P. lactiflora*. Geranyldiphosphate synthases and farnesyldiphosphate synthase catalyze the important branch-point reactions from GPP to monoterpenes and sesquiterpenes. The catalytic activities of the two enzymes are sensitive to temperature and metal ion concentration (*Kulkarni et al., 2013*). Under high temperature and $Mg^{2+}$ rich condition, the monoterpene concentration was significant higher compared with those of sesquiterpenes. In our experiment, we identified several unigenes encoding Geranyldiphosphate synthases and farnesyldiphosphate synthase in *P. lactiflora*. Their catalytic activities could be investigated under different temperature and metal ion conditions in follow-up studies.

It has been known that *P. lactiflora* is highly sensitive to the photoperiod and temperature changes. The accessions grown in the Bozhou region have been undergoing reproductive isolation for a long period. Thus, the genetic background of each individual might be fixed. In our study, we identified a large number of genes in *P. lactiflora* and found gene expression patterns in the terpenoid pathway are highly diversified among different accessions. We found The different biosynthetic abilities of terpenoid biosynthesis among the strains may be largely contributed to by several master genes, suggesting that the heterosis in crosses of PB and XT should be considered as a potential approach to breed *P. lactiflora* strains with high paeoniflorin levels. With the sequences of these genes, the phylogenetic structures and population genetics backgrounds of germplasm resources could be further investigated to elucidate the domestication and selection process of *P. lactiflora*.

## CONCLUSION

Using the RNA-sequencing protocol, we assembled the exon structures of 36,264 unigenes in the shoots and roots of three 3-year-old *P. lactiflorais* accessions, including 28,925 differentially expressed genes. We systematically annotated their molecular functions by aligning their sequences with those from multiple data resources. We found 3,952 genes with the Pearson Correlation Coefficients higher than 0.6 between the expression levels and the paeoniflorin levels suggesting their putative functions associated with paeoniflorin biosynthesis. We identified 32 genes encoding the enzymes controlling the major catalytic reactions in MVA and MEP pathways and one gene encoding (−)-alpha-terpineol synthase. These genes contributed nearly complete terpenoid backbone biosynthesis pathways. By profiling gene expression patterns associated with MVP, MEP pathways, and I-PP to monoterpene conversation reactions, we uncovered *AACT1, HMGS, MK, PMK, GGR, FPS1* and *MVD* that were highly expressed in the roots of high-paeoniflorin accessions, indicating that the different biosynthetic abilities of terpenoid biosynthesis among the accessions may be largely contributed to by several master genes.

## Accession numbers

The fastq-formatted RNA-seq datasets and the sampling information (accession number CRA001881) are publically available on the Genome Sequence Archive database (*BIG Data Center Members, 2018*; *Wang et al., 2017*).

### Funding

This work was supported by Province Science and Technology Major Project of Anhui (18030701163), University Natural Science Research Project in Anhui Province (KJ2016A490), Anhui Provincial Quality Engineering Projects in Colleges and Universities (2016sjjd054, 2016sxzx027) and Bozhou Action Plan for Leading Talents in Innovation and Entrepreneurship. The funders had no role in study design, data collection and analysis, decision to publish, or preparation of the manuscript.

### Grant Disclosures

The following grant information was disclosed by the authors:
Province Science and Technology Major Project of Anhui: 18030701163.
University Natural Science Research Project in Anhui Province: KJ2016A490.
Anhui Provincial Quality Engineering Projects in Colleges and Universities: 2016sjjd054 and 2016sxzx027.
Bozhou Action Plan for Leading Talents in Innovation and Entrepreneurship.

### Competing Interests

The authors declare that they have no competing interests.

### Author Contributions

- Baowei Lu conceived and designed the experiments, performed the experiments, prepared figures and/or tables, authored or reviewed drafts of the paper, and approved the final draft.
- Fengxia An conceived and designed the experiments, analyzed the data, prepared figures and/or tables, authored or reviewed drafts of the paper, and approved the final draft.
- Liangjing Cao performed the experiments, prepared figures and/or tables, and approved the final draft.
- Qian Gao analyzed the data, prepared figures and/or tables, authored or reviewed drafts of the paper, and approved the final draft.
- Xuan Wang analyzed the data, prepared figures and/or tables, and approved the final draft.
- Yongjian Yang performed the experiments, prepared figures and/or tables, and approved the final draft.
- Pengming Liu performed the experiments, authored or reviewed drafts of the paper, and approved the final draft.

- Baoliang Yang performed the experiments, authored or reviewed drafts of the paper, and approved the final draft.
- Tong Chen conceived and designed the experiments, analyzed the data, prepared figures and/or tables, and approved the final draft.
- Xin-Chang Li analyzed the data, prepared figures and/or tables, and approved the final draft.
- Qinghua Chen performed the experiments, authored or reviewed drafts of the paper, and approved the final draft.
- Jun Liu conceived and designed the experiments, prepared figures and/or tables, authored or reviewed drafts of the paper, and approved the final draft.

## Data Availability

The fastq-formatted RNA-seq datasets and the sampling information are publicly available as the Genome Sequence Archive database (CRA001881) at the BIG Data Center (PRJCA001310).

## Supplemental Information

Supplemental information for this article can be found online at http://dx.doi.org/10.7717/peerj.8895#supplemental-information.

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
