# Peer review of "Comparative transcriptomics characterized the distinct biosynthetic abilities of terpenoid and paeoniflorin biosynthesis in herbaceous peony strains"

_PeerJ, doi:10.7717/peerj.8895_

## Round 0.1 · original submission · Major Revisions

Dear authors
your ms has been reviewed. The reviewers ask for a major revision. Be aware that the same reviewers will check your revision. Thus make sure that you address their arguments.
Kind regards
Michael Wink
AE

[]

Reviewer 1 ·

Basic reporting

Most of the papers' english is understandable, but phrase and paragraph structure needs to be improved as the information is often not clearly stated. In the material an methods section, the latter are not extensively well described, so the experiments performed are not clear for us to understand as it stands.

Ambigous number of genes are given throughout the article (In the abstract and first part of results: 36,264- further 36,203 (supposedly these are the annotated ones? it is not clear). In the supplemental Table 3 there are 36,267 unigenes.) . Please stick to one number.

Some important references are missing, for instance another transcriptome study in Paeony (Luo, J etal. 2017. Transcriptomic Analysis Reveals Transcription Factors Related to Leaf Anthocyanin Biosynthesis in Paeonia. Molecules, doi:10.3390/molecules22122186).

The structure is overall right, supplemental data are given extensively. Nevertheless, some of the main Figures might be moved to the supplementals as the relevance for the paper is not really clear.

Furthermore, the authors miss phrasing the hypothesis clearly, which would be something like :” we expected the root to contain the paeoniflorin biosynthesis genes highly expressed compared to shoots as observed in Yuan et al. 2013…”.

In the abstract they mention "preferentially expressed" genes, but they should talk about "differentially expressed" instead. The numbers are quite high, but I think this depends on the statistical threshhold used to classify the genes, as I described later.

Experimental design

The research is an extensive analysis with RNAseq data, which complements a previous study, Yuan et al. (2013) often cited by the authors, as this study identifies 19 of the terpene backbone biosynthesis genes from ESTs (probably 454 sequencing).
In the current study, the authors find 14 more genes than Yuan et al., 13 from the same pathway, shown in Figure 4, and another involved in the conversion from "GPP to alpha-terpineol" as they write in the paper but don't show in the Figure 4. I suggest showing this step in a Figure, as PeerJ is not a phytochemistry journal and might be read by non-experts as well.
It might be interesting to use the same nomenclature used for the enzymes as the one used by Yuan et al. (2013), also in the heatmap Fig. 6, as to render them comparable and to be able to distinguish the newly identified genes. It might also be nice to highlight somehow the novel genes in Figure 4. And to go into more detail of the novel genes in the paper and their relationship with the way of sampling- why were these not identified in the previous study even if the tissue sampling was more extensive, is it because other strains were analyzed?
The comparison of the transcriptomes among strains is also nos extensively discussed, event if their analysis shows (PCA) that the strains have significant differences, even if the tissues do account for most of the differences analyzed.

The investigation appears rigorous, but the lack of detail in the Materials and Methods section gives the impression of superficiality in the analysis. A couple of details need to be included in the statistical analysis, as to why a cutoff of p=0.1 is used, further they mention Fold change, and it's rather log10FC. They have biological replicates, which is seen in the supplemental tables and the PC, but they don't mention it clearly in the text.

Could they please show how the gels/expression profiles of the GAPDH reference genes? In Yuan et al. (2013) they used PIActin, why did they not use the same one?

Please indicate the estimated efficiency from the standard curves for each primer (supplemental table 1). This is needed to be able to use the delta-deltaCT quantification Method mentioned previously.
The authors should please describe if two- or three- step cycles (95C denaturation, 60C annealing, etc) for the qPCR.

Validity of the findings

The underlying data are all provided, but they appear not to have been submitted to the public databases yet, NCBI- This needs to be done before re-submitting and the accession numbers given instead of, or additionally to, attaching the data.

There are novel data, as new genes in the paeoniflorin biosynthesis pathway were identified, and multiple transcription factors. It's important to annotate TFs, but the authors miss to show which of these are correlated (co-expressed) with terpene- paeoniflorin biosynthesis, to round-up their hypothesis.

Reference to the other potential interesting hypothesis to test should be made at least in the end of the discussion (strain comparisons, annotation o transcription factors related to paeoniflorin biosynthesis, etc.)

In the conclusions the authors should go into the hypothesis "comparing differentially expressed genes among tissues" which they don't go back to.

Additional comments

Dear authors,

The presented work is of general interest to the phytochemistry public, and the novel genes identified are important completion of the picture from previous work on paeoniflorin biosynthesis.
I hope the comments previously made will help you to improve the paper.
I attach the corrected pdf, for individual changes to be considered as well.

best regards.

Annotated reviews are not available for download in order to protect the identity of reviewers who chose to remain anonymous.

Reviewer 2 ·

Basic reporting

The manuscript is very hard to decipher. The Introduction fails to explain where the biosynthesis takes place and why then root and shoot tissues are selected for this study and not others. Furthermore, there is no short and clear statement about what to expect in every strain/tissue with respect to the production of the substances and/or their biosynthesis. Supplementary figure 1 should be included and explained in the main text, since it helps understanding the whole experimental set up.

The bioinformatic analysis is not thoroughly described, so reproducing the results would not be possible.

The whole description of the differential analysis (From line 214 on) is not understandable. Suggestion: “For each strain, we compared shoots VS roots and extracted lists of root/shoot specific upregulated genes. We then found 322/886 genes that were upregulated in all three strains, regarding them as tissue specific.“ In line 217 correct please the number 332 to 322 or viceversa.

In line 141 : differentially expressed genes are defined to have FC >2, pval < 0.05 and FDR<0.1 On line 216: FDR < 0.05: Which is correct?

Figure 2A please fix the labels for XT roots since they are all marked as rep2 and explain why one XT Root replicate is grouped outside the rest of root and shoots replicates from XT and PB.

Experimental design

The experimental design involves two different tissues from 3 strains producing different amounts of the substances of interest and 3 biological replicates per condition. This allows differential expression analysis between conditions to determine candidate genes involved in biosynthesis. A discussion of the reasons why roots and shoots are the best tissues for this purpose is missing.

Validity of the findings

The bioinformatic analysis is not thoroughly described, so reproducing the results would not be possible. Furthermore, it is almost impossible to validate the results with the lacking description of the analysis and results:

1. The big amount of de novo transcripts call my attention: did the authors post-process the transcriptome after Trinity to remove redundancies (e.g. CD-Hit) and artifacts (lowly expressed transcripts in all samples).

2. In order to be able to validate the assembled transcriptome, some important measures and their comparison to those in published transcriptomes should be shown and discussed. E.g. transcriptome statistics (N50, etc), transcript length distribution (Fig 3A), BUSCO completeness scores, etc
2.1 How were the reads aligned to the transcriptome (name of aligner and parameters), how were the reads counted to make a counts matrix for DESeq2, did the authors account for multiple alignments? How was the mapping percentage of raw reads to de-novo transcriptome?

3. The whole description of the differential analysis (From line 214 on) is not understandable. Suggestion: “For each strain, we compared shoots VS roots and extracted lists of root/shoot specific upregulated genes. We then found 322/886 genes that were upregulated in all three strains, regarding them as tissue specific.“ In line 217 correct the number 332 to 322 or viceversa, it doesn't match Figure 4.

4. In line 141 : differentially expressed genes are defined to have FC >2, pval < 0.05 and FDR<0.1 On line 216: FDR <0.05: Which is correct?

5. The gene lists discussed from line 214 on are not clearly explained. Not all of the found genes should be used in the next section. If no biosynthesis is expected on GS, I would remove from the candidate genes all those up-regulated in GS roots. If this does not make sense, please discuss.
5.1 The authors also show lists of genes up-regulated in shoots. From supp. figure 1, there is almost no substance in shoots of any strain. Since it is not clearly explained which genes were used from line 222 on, I assume you used all the up-regulated genes and that would be counter-intuitive. Please, explain which genes you used for enrichment analysis in the identification of genes in biosynthesis pathway (from line 222 on) , e.g. 'We used total of 2597 (XT-root) + 677 (XT and PB-root ) + 3737 (PB-root) for identifying candidate genes in the biosynthesis of ...'

6. The qPCR are ot shown in the main text but in supp. figure 2. The conclusion in lines 250-251 does not explain the fact , that except for AACT1, the rest genes don't seem to prove the hypothesis (if this is it!) that biosynthesis is occurring in roots and not in shoots and in XT, PB and not in GS. This qPCR results, instead of showing what one would expect, show almost the opposite. Otherwise, state please clearly what is the hypothesis.

7. Figure 6 is also not very useful the way it is. I would suggest use a color mapping red-green for low-high expression, which is standard and group columns according to the pattern we would like to see. Or plot log2FC, maybe will be clearer.

Additional comments

The authors have sequenced and de-novo assembled a reference transcriptome for the study of the biosynthesis of terpenoid and paeoniflorin in 3 different peony strains. The experimental design involves two different tissues from 3 strains producing different amounts of the afore mentioned substances and 3 biological replicates per condition. This allows differential expression analysis between conditions to determine candidate genes involved in biosynthesis. The work behind this paper has a great potential to serve as reference for further studies in this area.

However, the manuscript is not written in an understandable way, since no clear hypothesis is made, the analysis is not well described and thus not reproducible, the transcriptome is not backed up with a good validation.

I have big concerns about the validity of the candidate genes, since not even qPCR or expression heatmaps show a reasonable trend. I also doubt about the enrichment tests. If all genes found as differentially expressed (about 23K) were used for enrichment analysis, even by chance one could at a high probability choose 32 Unigenes related to biosynthesis.

---

## Round 0.2 · Major Revisions

Dear authors

Please make sure that you follow the advice of the reviewer. Otherwise we have to reject your ms.

Kind regards,

Michael Wink
Editor

Reviewer 1 ·

Basic reporting

I still had problems deciphering many of the phrases. I indicate most of the problems I had in an additional document.

Same with the references, I indicated one to improve their manuscript. Although missing ones in the previous version were included this time.

The structure is still not professional. I indicate the problems in the additional document.

The hypothesis were stated a bit clearer this 2nd time, but it still lacks a better connection among the strains and tissue analysis, which needs to be stated in the abstract and conclusions.

Experimental design

I think the research is very interesting, therefore it is crucial to make it more accesible to the public with the corrections needed.

Research question was better stated with re-submission, can still be improved.

Technical standards seem to have been followed, with exceptions, in the qPCR. The number of upregulated genes is still high, clarification is needed in the analysis. A higher cut-off might be needed for FDR, 0.01.

Methods need to be described better, e.g. Tissue dissection scheme, primer efficiencies in qPCR.

Validity of the findings

The results maybe novel. The significance, for the hypothesis, of the detected genes is not clear as it stands.

Data were made publicly available this time.

Conclusions need to be restated. Still many unclear points, for instance the number of transcription factors, the numbers of unigenes and differential expressed genes, which are not cited in either the main text or the Figures.

Additional comments

Dear authors,
Your study and data are very interesting and highly relevant for the lay public. Nevertheless, I have a complete review with still many issues to fix, so that the readers can get the crucial information and publication standards are met.

Best regards.

Complete review follows as an attached document.

Annotated reviews are not available for download in order to protect the identity of reviewers who chose to remain anonymous.

---

## Round 0.3 · Major Revisions

Dear authors

This will be your last chance to revise your ms. If you do not follow the recommendations of the reviewer, we have to reject your ms.

Regards
Michael Wink
Academic Editor

Reviewer 1 ·

Basic reporting

It is clear and unambigous. English is professional.

Background and Literature are good.

Ok, more comments later to improve paper readability.

Results are relevant to the hypothesis.

Experimental design

Correct.

Validity of the findings

Valid findings for an explorative study.

Additional comments

Dear authors,

This is a very interesting explorative study on the paeoniflorin biosynthesis in three Paeony strains.
I have a few corrections to point out to improve paper readability and/or quality.

L. 82-84. Please provide the abbreviations for the enzymes in the MVA pathway that will be used later in the text. This will help the reader to identify them quicker later, i.e. acetyl-CoA acetyltransferase (AACT1), hydroxyl methylglutaryl-CoA synthase (HMS), HMG-CoA reductase (HGMR) etc.

Correction in Figure quality, Fig3. Labelling of Subfigure C is not visible. Subfigure B, Correct to SwissProt, which is the correct name of the database (also in the text please!).

Figure 5: IMO: Enzyme name (EC:...) and Unigene Name should go in separate table. Else it will be too small to be read in a publication.

Other corrections:
L. 64 the correct name is of the deaseae is “typhoid fever”
L. 277-279: “These results ...” should be moved to Discussion section, as this is a very important speculation about the results.
L. 284 space missing: Spartina alterniflora

Else, it might be worth in a future study to determine the ploidy level of Paeony, as many transcripts are found in more than one copy, which might be an indication of polyploidy.

Best wishes!

---

## Round 0.4 · accepted · Accept

Dear authors

Congratulations- your revision is accepted, as it follows the recommendations of the reviewers.

Regards
Michael Wink
Academic Editor

Reviewer 1 ·

Basic reporting

Everything is ok.

Experimental design

Well explained.

Validity of the findings

All is well.

Additional comments

Dear authors,

After a careful review, I remarked that all previous concerns have been well addressed.
I suggest just a careful proofreading for spelling mistakes, but nothing critical anymore.